# Delayed Formation of Neonatal Reflexes and of Locomotor Skills Is Associated with Poor Maternal Behavior in OXYS Rats Prone to Alzheimer’s Disease-like Pathology

**DOI:** 10.3390/biomedicines10112910

**Published:** 2022-11-12

**Authors:** Tatiana Kozlova, Ekaterina Rudnitskaya, Alena Burnyasheva, Natalia Stefanova, Daniil Peunov, Nataliya Kolosova

**Affiliations:** 1Institute of Cytology and Genetics, Siberian Branch of Russian Academy of Sciences (ICG SB RAS), 10 Akad. Lavrentyeva Ave., 630090 Novosibirsk, Russia; 2Faculty of Natural Sciences, Novosibirsk State University, 2 Pirogova Str., 630090 Novosibirsk, Russia

**Keywords:** maternal behavior, early ontogenesis, unconditioned reflex, motor skill, Alzheimer’s disease, OXYS rat

## Abstract

Postnatal brain development is characterized by high plasticity with critical windows of opportunity where any intervention may positively or adversely influence postnatal growth and lead to long-lasting consequences later in life. Poor maternal care is among these interventions. Here, we found that senescence-accelerated OXYS rats prone to an Alzheimer’s disease-like pathology are characterized by more passive maternal behavior and insufficient care for pups as compared to control (Wistar) rats. OXYS pups demonstrated a delay in physical development (of auricle detachment, of emergence of pelage and incisors, of eye opening, and of vaginal opening in females) and late manifestation of reflexes and locomotor skills. All observed behavioral abnormalities are connected either with poor coordination of limbs’ movements or with a decrease in motivation and development of depression-like behavior. It is possible that their manifestations can be promoted by the features of maternal behavior of OXYS rats. Overall, these early-life events may have long-lasting consequences and contribute to neurodegeneration and development of the Alzheimer’s disease-like pathology later in life.

## 1. Introduction

The early postnatal period is critically important for the development of a mammalian brain. Growth and proper development during this period are the result of a combined interaction of complex functional, structural, and metabolic mechanisms, even minor changes in which can have far-reaching consequences that cannot be predicted at present. Nonetheless, the early postnatal period is characterized by high plasticity with critical windows of opportunity where any intervention may positively or adversely influence postnatal growth and development [1,2,3,4]. Furthermore, such an intervention may lead to long-lasting consequences later in life, e.g., inappropriate plasticity and even neurodegeneration [5,6].

At birth, a rodent body is not fully developed; therefore, maternal care is critically important in the first week of life. Indeed, periodic maternal deprivation from postnatal day (PND) 1 to PND6 has long-term consequences for the growth of developing rats: a decrease in body weight and an increased response to new stressors [7]. One of the body’s systems most vulnerable to external factors is the central nervous system, which controls the workings of the body as a whole. There is good evidence that maternal licking and grooming have a strong impact on the development of emotional and cognitive behavior later in life [8,9]. Natural differences in maternal care among rats have been shown to predict subsequent changes in spatial and emotional learning and memory, anxiety, social behavior, exploratory activity, and stress reactivity [10,11,12,13,14,15]. Moreover, a larger amount of maternal care early in life is associated with a decrease in age-dependent cognitive impairment [16]. Maternal care affects synaptogenesis in the hippocampus, GABA receptor expression, synaptic plasticity in the prefrontal cortex, and dendrite branching. In the early postnatal period, stress (namely, restriction of nest material) causes an increase in the formation of amyloid plaques in the hippocampus and a neuroinflammatory response to amyloid in old APP/PS1 mice [17]. Moreover, maternal care in rats, as an early-life experience, affects the whole life by means of the marks that it leaves in the epigenome of neural cells, thereby altering gene expression [18,19].

As we have shown earlier, accelerated aging of the brain and the development of signs of Alzheimer’s disease (AD) in senescence-accelerated OXYS rats can be attributed to delays in the completion of postnatal brain maturation [20,21,22]. Recently, our colleagues reported that replacement of senescence-accelerated OXYS rats’ mothers by recipient wild-type mothers leads to faster maturation of some neonatal reflexes and a change in the density of neurons in the hippocampi of offspring [23]. These results naturally raised a question: is the early development of neurodegenerative changes in OXYS rats connected with specific features of maternal behavior?

The purpose of this study was to compare the maternal behavior between OXYS and Wistar (control) rats and the dynamics of the development of reflexes as an indicator of the maturing of brain functions.

## 2. Materials and Methods

### 2.1. Animals

The rats were provided by the Breeding Experimental Animal Laboratory of the ICG SB RAS (Novosibirsk, Russia). The animals were kept under standard laboratory conditions (22 ± 2 °C, 60% relative humidity, and 12 h light/12 h dark cycle; the light was turned on at 8 a.m.) and had *ad libitum* access to standard rodent feed (PK-120-1, Laboratorsnab Ltd., Moscow, Russia) and water. We used adult female OXYS and Wistar rats (mean weight ~300 g) for analysis of a maternal behavior, OXYS and Wistar pups at PND1–PND20 (weight: from 4 g at PND0 to 45 g at PND20) for analysis of the behavior of pups, and OXYS and Wistar adolescents (PND45; weight 160–200 g) for assaying depression-like behavior. In each cage, only one female rat with its offspring was kept. When pups reached the age of PND20, they were separated from mothers and were kept as five sex-matched companions per cage.

Every effort was made to minimize the suffering of the animals: female rats were separated from their offspring for no more than 15 min daily; special lamps and heaters were used to prevent the cooling of the pups; the nests in the home cages were not disturbed when not necessary. Less than 5% of pups and no adult animals died during the experiments.

During the experiments, no randomization was performed. We did not exclude any animal from the analyses. For calculating group size, literature data were analyzed, and the needed number of animals (meaning necessary and sufficient numbers to carry out the experiment) was estimated.

### 2.2. Analysis of Maternal Behavior

Virgin 3-month-old female OXYS and Wistar rats (*n* = 4) were mated with males of the same strain and age. After that, the males were removed, and the females were kept in individual home cages. Videotaping of each cage was carried out following the birth of the pups. Daily activity of the animals was analyzed at 2 a.m., 8 a.m., 2 p.m., and 8 p.m. manually by an observer; then, the activity was averaged by hour. The following types of activity were examined: taking care of pups (feeding and grooming), taking care of themselves (feeding and grooming), locomotor activity (running and digging), and resting. Additionally, the overall time spent in the nest was determined.

### 2.3. Behavioral Analysis of the Pups

The scheme of the experiment is presented in Table 1 and Figure 1.

A modified version [24] of the open field test was used to evaluate the development of locomotion (which means raising the head and raising and coordinated movement of limbs) as well as the levels of locomotor activity (crawling, walking, and pivoting) and exploratory activity (rearing with and without support). The length of the open-field arena was 30 cm, and the width 20 cm; each animal was allowed to explore the arena for 3 min.

Analysis of righting in mid-air was conducted as described by Altman and Sudarshan [25]. Rats were dropped upside-down from a height of 60 cm onto a padded surface. Each animal was given three trials daily; two rightings out of three trials were regarded as successful acquisition of the ability.

The development of two types of placing reaction was investigated too, according to [25]. The first type is the extension and adduction of forelimbs with extension of the head, as seen in animals jumping down before landing. This reaction was elicited by suspending pups by their tails and then touching their vibrissae with a needle. The second type of placing reaction is the raising of shoulders and flexion of forelimbs, as seen in animals climbing an elevated object. This type of reaction was elicited by holding animals by the napes of their necks and then touching their chins with a horizontally oriented needle.

After that, we analyzed the ability of rats to overcome a gap between an empty cage and home cage by means of homing as motivation [25]. The initial gap between the cages was 1.2 cm. When the majority of the animals successfully passed this gap, the distance was extended to 3.8 cm.

To examine jumping abilities, each animal was placed on a pedestal situated above its home cage at heights of 7.5, 20, 35, and 50 cm. Each animal was given a single trial in ascending order of height every day [25]. Homing was used as a motivation.

Next, by means of rope and a wooden rod, we estimated abilities of the pups to climb up and down according to [25] with modifications; both the rope and rod had a diameter of 1.5 cm and a length of 30 cm. Each animal was placed on the rope or rod 30 cm above the home cage in one of two starting postures—with the head turned up or head turned down—and had to climb down; homing was employed as the motivation for the rats. To evaluate the ability of the pups to climb, a lower end of the rope or rod was placed in cold water, and an animal was placed just above the water surface with the head turned up. A platform 30 cm above the water level was connected with the rope or rod. Each animal had to climb the rope or rod to escape the cold water; a sibling placed on the escape platform served as an additional motivation.

Recognition memory was analyzed as described by Krüger and coauthors [26] with only one modification: because of later eye-opening in OXYS rats, we started to test memory at PND16 to be sure all the animals opened their eyes on the testing day. In brief, from PND12 to PND15, habituation trials were administered in which each rat was allowed to freely explore the arena and two identical objects for 10 min. These objects were not used during memory tests on PND16–PND18. At PND16, we tested novel object recognition (NOR) memory: first, there was 10 min of a familiarization trial, during which a rat was allowed to freely explore the arena and two identical objects; then, the animal was placed into a temporary holding cage for 10 min; finally, a randomly chosen familiar object was replaced by a novel object with different shape, color, and texture, and the animal was allowed to investigate the arena for 10 min. The object location recognition (OLR) task was held at PND17. First, there was a familiarization trial similar to that in the NOR test, followed by a test trial in which the objects did not change, but the position of one of the objects did. The recency recognition (RR) task was carried out at PND18. Two 10-min familiarization trials with two different pairs of objects were completed by the animal and were separated by an interval of 30 min. During the testing trial, which lasted for 5 min, one object used in the first familiarization trial and one object used in the second, more recent, familiarization trial were placed in the arena at the same positions as during the familiarization trials.

The salience of depression-like behavior was assessed in OXYS and Wistar rats at PND45 (*n* = 8) by the forced swim test [27], based on the procedure described in [28]. The run was being separated into two sessions, namely, a pre-test and test, 24 h apart. During the pre-test part, rats were individually placed in a glass cylinder 60 cm high and 30 cm in diameter containing water up to 40 cm at 25 °C. After 15 min of swimming, the rats were moved to a warmed drying cage and next to a home cage. The test session was administered under the same conditions for 5 min, accompanied by filming of the animals’ behavior. To evaluate depression-like behavior, behavioral patterns were assessed, such as time of immobility (only the movements that are necessary to stay above the water are present), struggling/climbing (intensive movements of the forelimbs leading to a rise of front paws above the water), and swimming (calm swimming with no attempts to escape). Among them, immobility is considered the most important in terms of representing the salience of depression-like behavior [27,28].

The timing of the following phenotypic parameters of all pups taking part in the experiments was recorded: auricle detachment, emergence of pelage, emergence of incisors, and eye opening. In addition, testis descent in males and vaginal opening in females were examined in offspring of four OXYS female rats and of four Wistar female rats.

Each behavioral test was recorded on a webcam and analyzed by an observer manually or in the Behavioral Observation Research Interactive Software (BORIS), v.8.0.9 (Torino, Italy). The observer was unaware of the animals’ group affiliation during the analysis of the videos.

### 2.4. Statistics

Statistical analysis was performed in STATISTICA 8.0 software (TIBCO Software Inc., Palo Alto, CA, USA). The Kolmogorov–Smirnov test was used to evaluate the normality of the data. Z-scores of ± 3 were utilized to detect outliers. The data were subjected to two-way analysis of variance (ANOVA). The genotype (i.e., strain) and sex were chosen as independent variables. Tukey’s post hoc test was applied to significant main effects and interactions in order to assess differences between some sets of means; the *t* test was performed to analyze differences between the strains; the *t* test for dependent samples was employed for a comparison of parameters within an animal; the Fisher exact test was used to assess the percentage of rats. The data are presented as the mean ± standard error of the mean (SEM). Differences were considered statistically significant at *p* < 0.05.

## 3. Results

### 3.1. Maternal Behavior

Firstly, we determined the time that rats spent in the nest. OXYS rats tended to spend more time in the nest as compared to Wistar rats (*t* test: *p* = 0.062): Wistar rats were in the nest for 40 ± 3 min per hour, whereas OXYS rats were in the nest for 47 ± 2 min/h. Seemingly, this increase was explained by a greater—almost twofold—amount of time spent on resting by OXYS rats (*t* test: *p* < 0.04). In addition, OXYS rats spent less time taking care of themselves (*t* test: *p* < 0.05) as compared to Wistar rats (Figure 2). Time spent by females on locomotor activity was the same between the strains. It is important to point out that, among OXYS rats, only one female of four devoted enough time to her pups (feeding and grooming), whereas all four female Wistar rats spent an appropriate amount of time feeding and grooming their pups.

### 3.2. Physical Development of the Pups

Manifestations of phenotypic parameters of rat development, such as auricle detachment and emergence of pelage and incisors (Table 2) were found to be delayed in OXYS rats (factorial ANOVA: F_1,275_ = 375.9, *p* < 0.0001; F_1,275_ = 51.9, *p* < 0.0001; and F_1,274_ = 12.6, *p* < 0.001, respectively). In addition, a comparison of group averages revealed that female OXYS rats open their eyes earlier than males do (*p* < 0.008), but still later than Wistar rats do. Furthermore, we showed a delay in vaginal opening in female OXYS rats (*p* < 0.001) and insignificantly delayed descent of testes in male OXYS rats (*p* = 0.07) as compared to Wistar rats (Table 2).

### 3.3. Development of the Quadruped Stance and Motion Coordination

Wistar and OXYS rats began to raise their heads above the surface and hold them in the raised state already on PND4. At the same time, a significantly smaller number of OXYS rats raised their heads on that day (Tukey’s test: *p* < 0.03). ANOVA did not reveal significant effects of either the genotype or sex on the mean day of raising and holding of the head by the animals (F_1,35_ = 1.4, *p* = 0.25; F_1,35_ = 0.001, *p* = 0.97, respectively). Nonetheless, the results of Tukey’s test indicated that on average, OXYS rats raised the head above the surface for the first time later than Wistar rats (*p* < 0.003, Table 3). All Wistar rats raised and held the head at PND8, and all OXYS rats could do it at PND9.

Rats of both strains began to raise their limbs on PND1. ANOVA did not detect any effects of the genotype and sex on the mean day of the raising of forelimbs (F_1,35_ = 1.1, *p* = 0.30; F_1,35_ = 0.008, *p* = 0.93, respectively). In contrast, we observed an effect of the genotype on the mean day of the raising of hindlimbs (F_1,35_ = 4.3, *p* < 0.05): indeed, OXYS rats began to raise their hindlimbs earlier than Wistar rats did (Table 3). Despite the earlier raising of the limbs, OXYS rats started to coordinate movements of their limbs later than Wistar rats. On average, the coordinated movements of forelimbs and hindlimbs developed later in OXYS rats (ANOVA: a “genotype” effect (F_1,35_ = 31.3, *p* < 0.0001) for forelimbs; marginal significance of a “genotype” effect (F_1,35_ = 4.0, *p* = 0.052) for hindlimbs).

Taken together, these results pointed to a delay in raising the head and coordinated movements of limbs in OXYS pups.

### 3.4. Development of Neonatal Reflexes in the Pups

Previously, we have demonstrated that the formation of the righting-on-surface reflex is delayed in OXYS pups [20]. Here, we evaluated the acquisition of the capacity for mid-air righting in OXYS and Wistar pups (Table 4). There were no effects of factors “genotype” and “sex” on the mean day of reflex formation (F_1,76_ = 0.7, *p* = 0.39; F_1,76_ = 0.6, *p* = 0.44, respectively). Indeed, on average, pups of both strains began to right their position in mid-air at PND12; 100% of Wistar rats demonstrated righting in mid-air at PND16, and all OXYS rats did so at PND17.

Next, we analyzed the placing reactions of the animals (Table 4). ANOVA did not reveal significant effects of factors “genotype” and “sex” on the mean day of emergence of the placing reaction seen in animals jumping down before landing (F_1,62_ = 2.7, *p* = 0.10; F_1,62_ = 1.5, *p* = 0.23, respectively). Indeed, rats of both strains started to demonstrate this placing reaction at PND1, and it appeared in all animals at PND10. The mean day of emergence of the placing reaction seen in animals jumping down before landing was PND5 in Wistar pups and PND6 in OXYS pups. Regarding another type of placing reaction, seen when an animal climbs an elevated object, ANOVA detected a strong effect of the genotype on the mean day of its emergence (F_1,62_ = 74.1, *p* < 0.0001). Indeed, although the emergence of this reaction was registered at PND1 in both rat strains, Wistar rats on average manifested the placing reaction—seen when an animal climbs an elevated object—at PND2, whereas in OXYS rats, this was observed at PND4 (Tukey’s test: *p* < 0.0001). Furthermore, 100% of Wistar rats demonstrated such a reaction at PND4, whereas all OXYS rats did so only at PND7.

Thus, on average, the placing reactions develop later in OXYS pups.

### 3.5. Locomotor and Exploratory Activities of the Pups

We started to evaluate locomotor and exploratory activity since PND11, when limb movements were found to be coordinated in all the animals.

#### 3.5.1. Locomotor Activity

At PND11, less than 10% of animals of both strains remained motionless. Indeed, at PND11, 94% of Wistar rats and more than 95% of OXYS rats crawled average distances of 36.0 and 55.6 cm, respectively (the difference is insignificant). Additionally, at PND11, 61% of Wistar rats and only 18% of OXYS rats started to walk keeping a raised quadruped posture with the entire body elevated (Fisher’s exact test, *p* < 0.0007). The average distance of walking did not differ between the strains: it was 47.7 cm for Wistar rats and 41.0 cm for OXYS rats. The distance of neither crawling nor walking was affected by the sex of the animals at PND11. The distance crawled naturally decreased from PND11 to PND20 in both rat strains (*t* test for dependent samples: *p* < 0.0002 for Wistar rats, *p* < 0.0001 for OXYS rats) with simultaneous extension of the walking distance (*t* test for dependent samples: *p* < 0.0002 for Wistar rats, *p* < 0.05 for OXYS rats). Furthermore, we revealed different dynamics of walking-distance changes between Wistar and OXYS rats (Figure 3A). Namely, in Wistar rats, the walking distance remained relatively unchanged from PND11 to PND15 and slightly lengthened by PND16 (*t* test (dependent samples) for a difference between PND11 and PND16: *p* < 0.05). Then, the parameter went up 2.5-fold until PND17 (*t* test (dependent samples) for a difference between PND16 and PND17: *p* < 0.002), decreased by PND18 (*t* test (dependent samples) for a difference between PND17 and PND18: *p* < 0.05), increased again by PND19 (*t* test (dependent samples) for a difference between PND18 and PND19: *p* < 0.0001), and remained at this level at PND20. By contrast, in OXYS rats, the walking distance gradually increased starting from PND11 (*t* test for dependent samples: *p* < 0.008 for a difference between PND12 and PND13; *p* < 0.02 for a difference between PND13 and PND14; *p* < 0.03 for a difference between PND15 and PND16) reaching the maximum at PND17 and then declined 1.5-fold until PND20 (*t* test (dependent samples) for a difference between PND17 and PND20: *p* < 0.009). These different dynamics led to interstrain differences in the walking distance: indeed, this parameter in OXYS rats was greater relative to Wistar rats at PND13, PND14, PND15, PND16, and PND18 (Tukey’s test: *p* < 0.002, *p* < 0.02, *p* < 0.0002, *p* < 0.0002, and *p* < 0.03, respectively), whereas at PND19 and PND20, OXYS rats walked less than Wistar rats did (Tukey’s test: *p* < 0.006 for PND19; only a trend at PND20: *p* = 0.059). Additionally, we found that at PND19 the walking distance was influenced by the sex of the animals (ANOVA: F_1,35_ = 4.6, *p* < 0.04); however, Tukey’s test showed that the effect was only significant in OXYS rats: female rats walked less than male rats (*p* < 0.03).

The number of pivots is another indicator of locomotor activity of animals. We observed a difference in the dynamics of the number of the pivots between the two strains (Figure 3B). In Wistar rats, the parameter slightly decreased by PND14 (*t* test (dependent samples) for a difference between PND11 and PND14: *p* < 0.01) and then rose until PND20 (*t* test (dependent samples) for a difference between PND14 and PND20: *p* < 0.0001). On the other hand, in OXYS rats, the pivot number increased by PND13 (*t* test (dependent samples) for a difference between PND11 and PND13: *p* < 0.003) and ultimately declined by PND20 (*t* test (dependent samples) for a difference between PND13 and PND20: *p* < 0.0001). Thus, the number of pivots was higher in OXYS rats relative to Wistar rats at PND11, PND12, PND13, PND14, PND15, PND16, and PND17 (Tukey’s test: *p* < 0.003, *p* < 0.0001, *p* < 0.0001, *p* < 0.0001, *p* < 0.004, *p* < 0.0003, and *p* < 0.01), reached the level of Wistar rats at PND18–19, and became lower at PND20 (Tukey’s test: *p* < 0.005).

#### 3.5.2. Exploratory Activity

The number of rearings (with or without support) reflects the exploratory activity of animals (Figure 4). In Wistar rats, the number of rearings went up from PND11 to PND17 (*t* test for dependent samples: *p* < 0.0001), then declined by PND18 (*t* test for dependent samples: *p* < 0.04) and increased again by PND19 (*t* test for dependent samples: *p* < 0.0003) remaining at the same level at PND20. On the one hand, the number of rearings rose in OXYS rats (as in Wistar rats) from PND11 to PND17 (*t* test for dependent samples: *p* < 0.0001); on the other hand, we did not notice a decrease of this parameter until PND18: instead, we detected it between PND18 and PND19 (*t* test for dependent samples: *p* < 0.004). Thus, OXYS rats carried out more rearings at PND11, PND12, PND13, PND14, PND15, and PND18 relative to Wistar rats (Tukey’s test: *p* < 0.007, *p* < 0.005, *p* < 0.0001, *p* < 0.0001, *p* < 0.0002, and *p* < 0.0006, respectively), but there was an almost significantly lower number of rearings in OXYS rats at PND19 and PND20 (Tukey’s test: *p* = 0.064 and *p* = 0.057, respectively). As for the rearing type, Wistar rats started to perform the more difficult rearing (without support) at PND15, whereas OXYS rats manifested this trait at PND13. OXYS rats carried out a larger number of rearings without support as compared to Wistar rats at PND15 (Tukey’s test: *p* < 0.005), and there was a somewhat lower number of rearings without support performed by OXYS rats at PND20 (Tukey’s test: *p* = 0.057). We did not observe the influence of the sex of the animals on the total number of rearings or on the number of rearings without support.

Thus, OXYS pups demonstrate higher locomotor and exploratory activities during the second decade of life; however, by the end of this period, their activity decreases, becoming lower than that in Wistar pups.

### 3.6. Jumping Abilities of the Pups

The ability of the animals to cross the gap between the empty cage and home cage was assessed starting from PND4 (Table 5). The initial width of the gap was 1.2 cm; when an animal succeeded in clearing this obstacle, the gap was widened to 3.8 cm. We did not find an influence of the factors “sex” (F_1,51_ = 0.9, *p* = 0.35) and “genotype” (F_1,51_ = 0.1, *p* = 0.76) on the postnatal day of crossing (stepping or jumping) the 1.2 cm gap by the animals. On average, Wistar and OXYS rats crossed the gap at PND9, with 100% of animals of both strains successfully crossing the 1.2 cm gap on PND13. Nevertheless, a different pattern was documented when the gap was widened to 3.8 cm. Indeed, more than 50% of Wistar rats and only 12% of OXYS rats cleared this obstacle by PND20 (ANOVA: F_1,53_ = 9.3, *p* < 0.004).

The ability to jump from a height was evaluated starting from PND5. Every day, the animals had to jump down to a home cage from a platform located at the increasing height: 7.5, 20, 35, and 50 cm. We did not detect any interstrain differences in the acquisition of the ability to jump from a height of 7.5 cm, but analysis of the data suggested that at PND5 and PND6, the percentage of pups that jumped from heights of 20, 35, and 50 cm depended on the genotype (PND5: F_1,76_ = 31.9, *p* < 0.0001 for 20 cm; F_1,76_ = 17.7, *p* < 0.0001 for 35 cm; F_1,76_ = 14.3, *p* < 0.0003 for 50 cm; PND6: F_1,76_ = 8.1, *p* < 0.006 for 20 cm, F_1,76_ = 104.1 *p* < 0.0001 for 35 cm; F_1,76_ = 260.3, *p* < 0.0001 for 50 cm). Indeed, at PND5, 90% of Wistar rats jumped from the height of 20 cm (Figure 5A), 80% of them jumped from the height of 35 cm (Figure 5B), and 57% from the height of 50 cm (Figure 5C), whereas only 38% of OXYS rats jumped from heights of 20 and 35 cm (Figure 5A,B), and 15% jumped from the height of 50 cm (Figure 5C). At PND6, more than 90% of Wistar rats jumped from all the proposed heights, whereas among OXYS rats, 80% jumped from the height of 20 cm, 20% jumped from the height of 35 cm, and only 8% from the height of 50 cm. Nonetheless, it should be pointed out that, by PND7, the interstrain differences disappeared. Indeed, starting from PND7 and until eye-opening, more than 90% of the animals of both strains jumped from all the offered heights (Figure 5A–C). As expected, the eye-opening led to a considerable change in rat behavior: at PND16, all OXYS rats jumped from the height of 20 cm, whereas at PND17, only 80% of these animals did (*t* test for dependent samples: *p* < 0.02). Among Wistar rats, the percentage of rats that jumped from the height of 50 cm on PND17 diminished > 1.5-fold as compared to the previous day (*t* test for dependent samples: *p* < 0.003). At PND19 and PND20, the percentage of OXYS rats that jumped from heights of 20 and 50 cm was significantly lower than that of Wistar rats (PND19: F_1,61_ = 14.3, *p* < 0.0004 for 20 cm; F_1,61_ = 4.6, *p* < 0.04 for 50 cm; PND20: F_1,61_ = 4.2, *p* < 0.05 for 20 cm; F_1,61_ = 9.0, *p* < 0.004 for 50 cm).

Additionally, we detected an effect of sex on the percentage of animals that jumped at PND17: the number of females that jumped from the height of 20 cm was lower (F_1,61_ = 4.6, *p* < 0.04) and the number of females that jumped from the height of 35 cm was higher (F_1,61_ = 4.6, *p* < 0.04) than that of males.

Therefore, Wistar and OXYS rats were equally successful in crossing the 1.2 cm gap, but a much smaller percentage of OXYS rats successfully cleared the 3.8 cm gap. The beginning of the formation of the vertical jumping ability in OXYS rats proved to be delayed. Starting from PND7, the differences from Wistar rats disappeared, but reappeared after PND19.

### 3.7. Abilities to Climb Up and Down

#### 3.7.1. The Ability to Climb Down

We analyzed the animals’ ability to climb down the rope or rod starting in one of two postures: with the head up or head down. We began by assessing the climbing down the rope in the head-up posture, which allows for rats to hold tight on the rope in order to avoid falls. Animals of both strains started to climb down successfully after PND10. By PND16, all Wistar rats successfully climbed down the rope, whereas all OXYS rats manifested this skill only at PND18 (Figure 6A). According to ANOVA, the average postnatal day of the skill formation was influenced only by the genotype (F_1,52_ = 22.6, *p* < 0.001). On average, Wistar rats started to climb down the rope in the head-up posture at PND13, whereas OXYS rats demonstrated this ability only at PND15 (Tukey’s test: *p* < 0.0001). Regarding the time required to climb down from the top of the rope in the head-up posture, this naturally increased after PND3, as animals demonstrated fewer falls and more successful attempts to climb down. In Wistar rats, this parameter reached the maximum on PND13—the mean day of their skill formation—and then declined by PND20 as the animals improved their climbing skill (*t* test (dependent samples) for a difference between PND13 and PND20: *p* < 0.0001). By contrast, in OXYS rats, the peak in the time needed to climb down was registered at PND14 and did not change until the end of the experiment; thus, this was higher at PND20 than that in Wistar rats (Tukey’s test: *p* < 0.002).

In addition, it should be mentioned that maturation of the climb-down pattern—meaning rotation of the trunk and climbing with the head in the leading position—started from PND13 in Wistar rats and from PND8 in OXYS rats; in contrast, all Wistar rats changed their climbing pattern by PND17, and all OXYS rats did so by PND18.

As already mentioned, climbing down in the head-down posture was assumed to be a mature pattern of climbing down; however, it was more difficult for the animals to not fall from the rope in this posture. Successful attempts to climb down the rope in the head-down posture appeared at PND11 in Wistar rats and at PND12 in OXYS rats. The mean day of the skill acquisition was influenced only by the genotype of the animals (ANOVA: F_1,52_ = 21.3, *p* < 0.0001) and was later in OXYS rats (Figure 6B). Regarding the time necessary for the animals to climb down in the head-down posture, it naturally increased simultaneously with an increase in the number of successful attempts to climb down and reached the maximum at PND14 in Wistar rats and at PND16 in OXYS rats. After that, the climbing time decreased until PND20 as the animals improved their ability to climb down, though the parameter remained higher in OXYS rats than in Wistar rats at PND20 (Tukey’s test: *p* < 0.0001).

Next, we investigated the ability of the animals to climb down the rod in head-up and head-down postures (Figure 6C,D). Wistar rats started to hold tight on the rod by PND3; OXYS rats did so by PND4. Wistar rats demonstrated successful climbing down the rod in the head-up posture starting from PND16, while OXYS rats started to climb down at PND14. By PND20, 100% of OXYS rats and 93% of Wistar rats climbed down the rod in the head-up posture (Figure 6C). By contrast, the mean day of the acquisition of the climb-down ability was influenced neither by the genotype (ANOVA: F_1,50_ = 1.1, *p* = 0.30) nor by sex (ANOVA: F_1,50_ = 0.02, *p* = 0.90) and was PND17.5. The time needed to climb down from the top of the rod in the head-up posture lengthened as the animals experienced fewer falls and made more successful attempts to climb down. This parameter was lower in OXYS rats at PND4, PND5, PND6, PND12, and PND13 (Tukey’s test: *p* < 0.01, *p* < 0.0004, *p* < 0.04, *p* < 0.005, and *p* < 0.0004, respectively); also, the parameter tended to be lower in OXYS rats at PND16 (Tukey’s test: *p* = 0.068). Then, the time needed for OXYS rats to climb down the rod reached the level of Wistar rats at PND17—the mean day of successful climbing down for both strains—and continued to grow until PND20 (*t* test (dependent samples) for a difference between PND17 and PND20: *p* < 0.008), thus becoming greater than that in Wistar rats at PND18, PND19, and PND20 (Tukey’s test: *p* < 0.0001, *p* <0.006, and *p* < 0.03, respectively).

The climbing down the rod in the head-down posture was much more difficult than that in the head-up posture (Figure 6D): indeed, by PND20, only 57% of Wistar rats and 73% of OXYS rats successfully climbed down (an insignificant difference: Fisher’s exact test, *p* = 0.16). Wistar rats started to hold tight on the rod at PND10, whereas OXYS rats did so following PND13. In OXYS rats compared to Wistar rats, the duration of staying on the rod without falling was shorter at PND13 and PND16 (Tukey’s test: *p* < 0.05 and *p* < 0.002, respectively) and was longer at PND18 and PND19 (Tukey’s test: *p* < 0.02 for both time points); at PND20, the time spent on the rod by OXYS rats tended to be shorter relative to Wistar rats (Tukey’s test: *p* = 0.068).

#### 3.7.2. The Ability to Climb

First, we analyzed the ability of the animals to climb the rope (Figure 7A). We saw that, by PND20, 93% of Wistar rats and 100% of OXYS rats successfully climbed the rope. ANOVA revealed that the mean day of successful climbing was influenced only by the genotype (F_1,50_ = 7.9, *p* < 0.007) and was almost 1 day later in OXYS rats. In Wistar rats, the average time needed to climb the rope lengthened from PND15 to PND16 (*t* test for dependent samples: *p* < 0.04) and then shortened to PND20 (*t* test (dependent samples) for a difference between PND16 and PND20: *p* < 0.001), thereby reflecting an improvement in the animals’ climbing skills. In OXYS rats, the peak in this parameter was registered at PND18, and time spent on the rope was more than twofold longer than that in Wistar rats (Tukey’s test: *p* < 0.0008).

Regarding the climbing of the rod (Figure 7B), Wistar rats started to climb up successfully after PND18, and by PND20, 30% of these animals climbed the rod, whereas only 4% of OXYS rats (a single animal) climbed up at PND20 (Fisher’s exact test, *p* < 0.02). The duration of staying on the rod without falling gradually increased from PND15 to PND20 in Wistar and OXYS rats (*t* test (dependent samples) for a difference between PND15 and PND20: *p* < 0.009 and *p* < 0.005, respectively). This parameter was lower in OXYS rats compared to Wistar rats at PND15, PND16, PND17 (insignificantly), and PND19 (Tukey’s test: *p* < 0.01, *p* < 0.006, *p* = 0.061, and *p* < 0.0002, respectively).

Thus, overall, the abilities to climb up and down arose later in OXYS pups than in Wistar pups.

### 3.8. Recognition Memory in the Pups

During the NOR task at PND16, we observed that the distance covered in the familiarization trial was shorter in OXYS pups (ANOVA: F_1,36_ = 4.3, *p* < 0.05), and there was an interaction between factors “genotype” and “sex” (ANOVA: F_1,36_ = 4.7, *p* < 0.04). Nonetheless, in the testing trial, the differences disappeared. Animals of both strains did not demonstrate a preference for one of the objects—namely, right or left—during the familiarization trial in the NOR task (Table 6). Wistar rats tended to spend more time investigating the novel object during the testing trial as compared to investigating the object in the same place during the familiarization trial (*t* test for dependent samples: *p* = 0.068), and OXYS rats spent significantly more time investigating the novel object (*t* test for dependent samples: *p* < 0.01). Of note, this was not the case for investigation of the familiar object: the time spent by rats of both strains did not differ between the familiar object during the testing trial and the object in the same place during the familiarization trial (*t* test for dependent samples: *p* > 0.05). Another important observation is that only male Wistar rats tended to spend more time investigating the novel objects (*t* test (dependent samples) for differences between familiarization and testing trials: *p* = 0.055), while female rats did not (*t* test (dependent samples) for differences between familiarization and testing trials: *p* = 0.78). Only female OXYS rats spent significantly more time investigating the novel object (*t* test (dependent samples) for differences between familiarization and testing trials: *p* < 0.02), but OXYS males did not (*t* test (dependent samples) for differences between familiarization and testing trials: *p* = 0.084). Additionally, the number of times the animals approached either a novel or familiar object was greater during the testing trial compared to the familiarization trial only in OXYS rats (*t* test for dependent samples: *p* < 0.01 for both objects). This result might be a consequence of significantly—almost 2.5-fold—longer distance traveled by OXYS rats during the testing trial as compared to the familiarization trial (*t* test for dependent samples: *p* < 0.001). Therefore, during the testing trial, despite the absence of differences between the novel and familiar objects in the rats’ investigation, animals of both strains spent more time investigating only the novel object in the testing trial compared to the object in the same place in the familiarization trial.

In the OLR task, we found that the distance covered by the animals during the familiarization trial was affected by the genotype (ANOVA: F_1,36_ = 10.1, *p* < 0.003) and sex (F_1,36_ = 4.4, *p* < 0.05), and there was an interaction between these factors (F_1,36_ = 7.8, *p* < 0.04). Indeed, the distance traveled by OXYS rats was shorter in comparison with Wistar rats (ANOVA: F_1,38_ = 8.5, *p* < 0.006); additionally, female Wistar rats covered more than a twofold greater distance as compared to males (Tukey’s test: *p* < 0.02). All these differences disappeared in the testing trial (Table 7): indeed, OXYS rats traveled much longer distance during the testing trial than during the familiarization trial (*t* test for dependent samples: *p* < 0.002); the same was true for males of the Wistar strain (*t* test for dependent samples: *p* < 0.02). We did not detect any differences in the interaction (meaning the time spent investigating the objects and approaching the objects) of rats of both strains with objects put in the old or new place. In addition, we found that animals of both strains spent more time near both the object put in the old place and the object put in the new place in the testing trial compared to the familiarization trial (*t* test for dependent samples: *p* < 0.04 for the object put in the new place for Wistar rats; *p* < 0.05 for the object put in the old place for Wistar rats; *p* < 0.004 for the object put in the new place for OXYS rats; *p* < 0.02 for the object put in the old place for OXYS rats); moreover, OXYS rats made more approaches to both objects in the testing trial than in the familiarization trial (*t* test for dependent samples: *p* < 0.006 for the object put in the new place; *p* < 0.01 for the object put in the old place). Again, it is important to emphasize that these differences in the interaction with objects between the testing and familiarization trials were characteristic of male rats of both strains, and not of females.

Regarding the RR task (Table 8), we demonstrated that OXYS rats covered twofold shorter distance during the testing trial as compared to Wistar rats (ANOVA: F_1,36_ = 9.4, *p* < 0.004). OXYS rats spent less time with the object that they explored during the first familiarization trial (ANOVA: F_1,36_ = 5.8, *p* < 0.02), whereas there was no genotype effect on the time spent investigating the object from the second familiarization trial (ANOVA: F_1,36_ = 3.2, *p* = 0.080). Regarding the number of approaches to the objects, this was less in OXYS rats for both objects (ANOVA: F_1,36_ = 8.1, *p* < 0.007 for the object from the first familiarization trial; F_1,36_ = 8.9, *p* < 0.005 for the object from the second familiarization trial) than in Wistar rats. Furthermore, we noticed that animals of both strains spent more than 1.5-fold more time with the object from the first familiarization trial; however, the differences were insignificant (*t* test for dependent samples: *p* > 0.05). Nevertheless, when sex differences were examined, we revealed that female Wistar rats spent significantly more time with the object that they explored during the first familiarization trial than with the object from the second familiarization trial (*t* test for dependent samples: *p* < 0.05).

Thus, examination of the learning ability of the rats revealed sex differences: only Wistar males and OXYS females distinguish a novel from familiar object; moreover, only Wistar females recognize the object in the RR test. Shifting the location of the familiar object stimulates the activity of both Wistar and OXYS rats.

### 3.9. The Forced Swim Test

This test was performed on OXYS and Wistar rats at the age of 45 days (Figure 8). The time of immobility—the main metric of the salience of depression-like behavior—was significantly longer in OXYS rats than in Wistar rats (*p* < 0.04).

## 4. Discussion

Here, we investigated the formation of neonatal reflexes and postural and locomotor skills in OXYS rats from birth to PND20. In this period, rat pups are highly dependent on maternal care [30]. We showed that female OXYS rats spend a lot of time in their nests for resting as compared to control (Wistar) rats. Additionally, only one OXYS female of four devoted enough time to its pups. Regarding Wistar rats, all four females spent an appropriate amount of time with their pups. The observed features of maternal behavior may influence the development of OXYS pups during the first postnatal week, which is considered the most sensitive period for the formation of brain structures. This supposition is in agreement with data from our colleagues: Igonina and coauthors [23] reported improved neurogenesis and normalized formation of some neonatal reflexes in OXYS pups with mothers replaced by wild-type female rats.

Features of early development of OXYS rats represent a delay in almost all examined parameters—auricle detachment, emergence of pelage and incisors, eye opening, and vaginal opening in females—with the exception of testis descent in males (an insignificant delay). It is important to emphasize that the majority of these parameters depend on apoptosis [31,32]; accordingly, their retardation may point to a delay in apoptosis waves in OXYS rats. This observation is in agreement with the results obtained earlier [21] concerning the delay in apoptosis in the OXYS rat brain.

Our examination of the development of the quadruped stance revealed that OXYS pups raise their head above the surface and start holding it in the raised position later than Wistar pups. Furthermore, coordinated limb movements were found to start forming later in OXYS rats compared to Wistar rats. Regarding the correct execution of the mid-air righting reflex, coordinated movements of the whole body are needed [33], and the delay in the formation of this reflex indicates poor movement coordination in OXYS pups. In addition, placing reactions are strongly related to coordinated limb movements. We noticed that the formation of the placing reaction during climbing an elevated object was delayed in OXYS rats, and there was an insignificantly delayed formation of the reaction during jumping down before landing. The most difficult skills that require well-coordinated movements of limbs are the abilities to climb up and down. To test these, we used rope and a wooden rod. We noted that both abilities develop later in OXYS rats than in Wistar rats. It is worth pointing out that in this study, Wistar rats improved their ability to climb down the rope and rod in the head-up posture, whereas OXYS rats did not. Climbing down the rope in the head-down posture is a more difficult skill than climbing down in the head-up posture [25]; hence, it formed later in rats of both strains. Both Wistar and OXYS rats exhibited an improvement in the ability to climb down in the head-down posture. Taken together, these findings clearly point to a retarded onset of coordinated limb movements in OXYS rats. It is interesting to note that Roberto and Brumely [34] have shown improved motor coordination during sensory-evoked motor responses in prematurely delivered rats compared to age-matched controls. Those authors explain this observation in terms of manipulating the amount of postnatal experience in perinatal rats: an increase in the parameter results in more mature movement patterns. Previously, we demonstrated that OXYS pups are born prematurely as compared to Wistar rats [35]; however, here, we claim a decay of movement coordination in OXYS rats, which may be explained by factors other than shortened gestation.

Locomotor activity in rodents strengthens during the first postnatal month [36]. Nevertheless, we documented a decline in locomotion at PND18 in Wistar rats; this drop seems similar to the decrease in locomotion in rats at PND18 reported by Altman and Sudarshan [25]. Initially, OXYS rats demonstrated a retardation of motor skill formation: indeed, the walking distance was shorter and the number of pivots was higher in OXYS rats, suggesting that OXYS pups pivoted in place rather than walking across the arena. This result again points to poor coordination of limb movements.

We started an examination of locomotion when pups of both strains were still blind (at PND11). It is expected that eye-opening will influence animals’ locomotion: Altman and Sudarshan [25] have reported a drastic increase in the distance traveled by rats after eye-opening. We did not observe this effect in Wistar rats; however, eye-opening led to greater locomotion in OXYS rats. After eye-opening, the number of pivots changed in opposite directions when rats of the two strains were compared; namely, in Wistar rats, this parameter went up, probably indicating the growth of exploratory activity following the opening of eyes, whereas in OXYS rats, the number of pivots diminished, implying an improvement of limbs’ coordination. The growth in exploratory activity in both rat strains manifested itself in a greater number of rearings after eye-opening. Another parameter influenced by the opening of eyes was the ability to jump from a height. Expectedly, eye-opening was followed by a decrease in the frequency of jumping; however, OXYS rats continued jumping less often than Wistar rats.

During our examination of the animals’ learning abilities (NOR, OLR, and RR tests), we noted that, in an open-field arena, the presence of different objects per se increased the locomotion of Wistar rats and did not influence this in OXYS rats. That is, although, in the open-field test, the distance covered by OXYS rats at PND16 and PND18 was longer relative to Wistar rats, during NOR, OLR, and RR tests, this parameter was lower in OXYS rats at PND16, PND17, and PND18. Nonetheless, it should be underscored that changing the objects or merely changing their position within the arena stimulated locomotor activity of OXYS rats: although OXYS rats traveled a shorter distance during habituation trials than Wistar rats, this difference disappeared during test trials. These results may be attributed to decreased motivation for exploratory activity and development of depression-like behavior, which we observed during the forced swim test in OXYS rats. These behavioral features may be consequences of poor maternal behavior. It is widely accepted that maternal separation during first 2–3 weeks of life leads to anxiety and depression-like behavior, as well as worsening motivation in young animals [37,38]. Only Wistar males and OXYS females were found to distinguish the novel from familiar object, and only Wistar females recognized the object in the RR test.

Here, we demonstrated the phenotypic outcomes of retardation of brain development in OXYS rats; this retardation was reported by us us earlier [21]. All behavioral abnormalities were connected either with poor coordination of limbs’ movements or with a decrease in motivation and development of depression-like behavior, or all of the above. This may be a result of insufficient maternal care in OXYS rats. These early-life events may have long-lasting consequences and contribute to neurodegeneration and the development of an AD-like pathology later in life. This possible link should be researched in further studies.

## Figures and Tables

**Figure 1 biomedicines-10-02910-f001:**
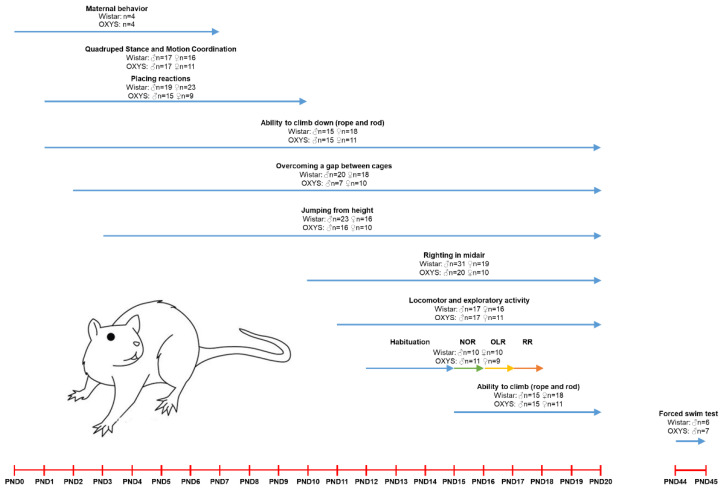
The scheme of the experiment.

**Figure 2 biomedicines-10-02910-f002:**
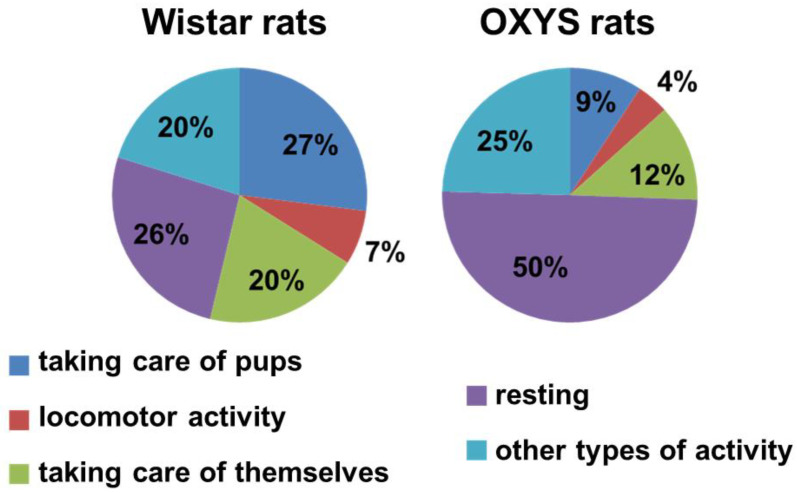
Maternal behavior of OXYS and Wistar rats.

**Figure 3 biomedicines-10-02910-f003:**
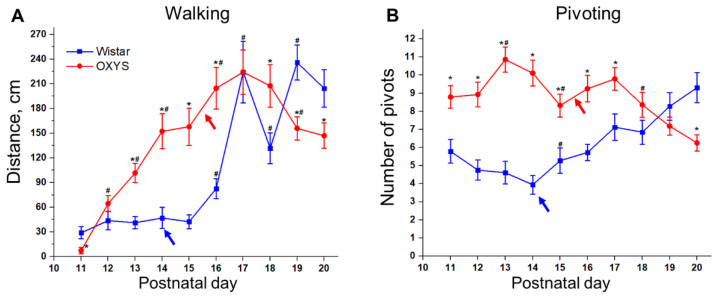
Locomotor activity of OXYS and Wistar pups. (**A**) The walking distance; (**B**) the number of pivots. * *p* < 0.05 as compared to control (Wistar) rats; ^#^
*p* < 0.05 as compared to a previous day; blue arrow: eye opening in Wistar pups; red arrow: eye opening in OXYS pups. The data are presented as the mean ± SEM.

**Figure 4 biomedicines-10-02910-f004:**
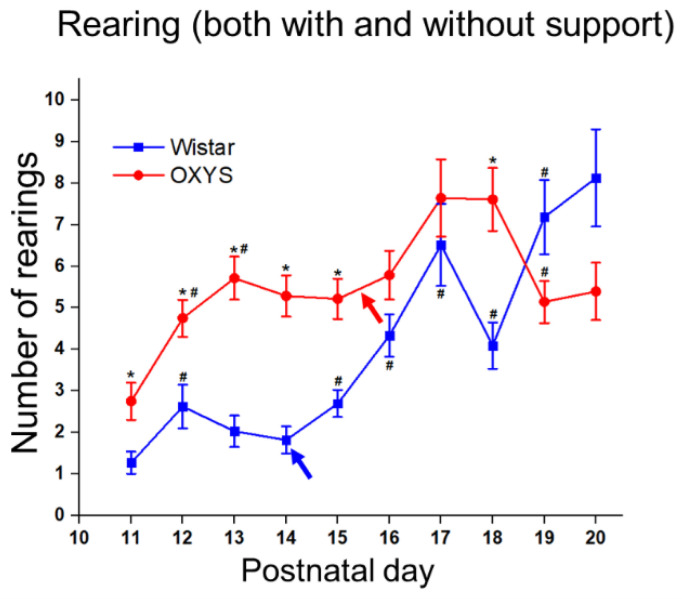
Exploratory activity of OXYS and Wistar pups. * *p* < 0.05 as compared to control (Wistar) rats; ^#^
*p* < 0.05 as compared to a previous day; blue arrow: eye opening in Wistar pups; red arrow: eye opening in OXYS pups. The data are presented as the mean ± SEM.

**Figure 5 biomedicines-10-02910-f005:**
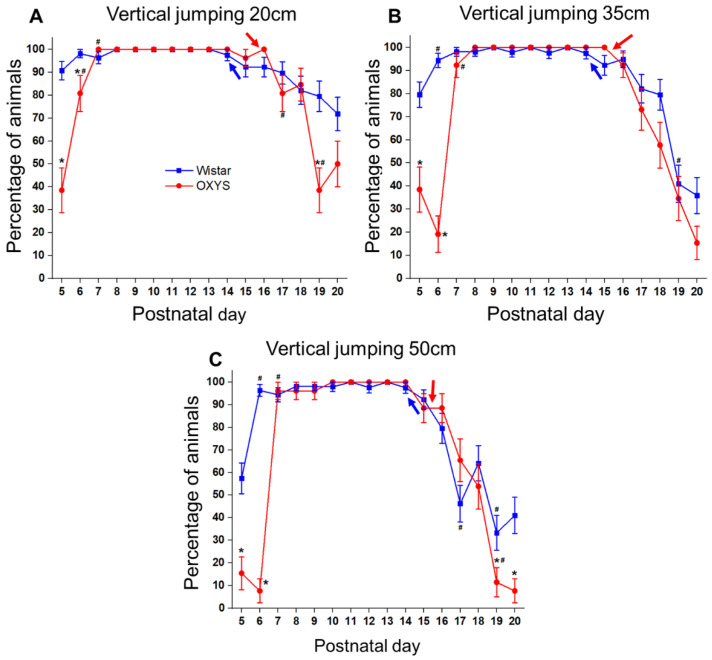
The ability of OXYS and Wistar pups to jump. (**A**) Jumping from a height of 20 cm; (**B**) jumping from a height of 35 cm; (**C**) jumping from a height of 50 cm. * *p* < 0.05 as compared to control (Wistar) rats; ^#^
*p* < 0.05 as compared to a previous day; blue arrow: eye-opening in Wistar pups; red arrow: eye opening in OXYS pups. The data are presented as the mean ± SEM.

**Figure 6 biomedicines-10-02910-f006:**
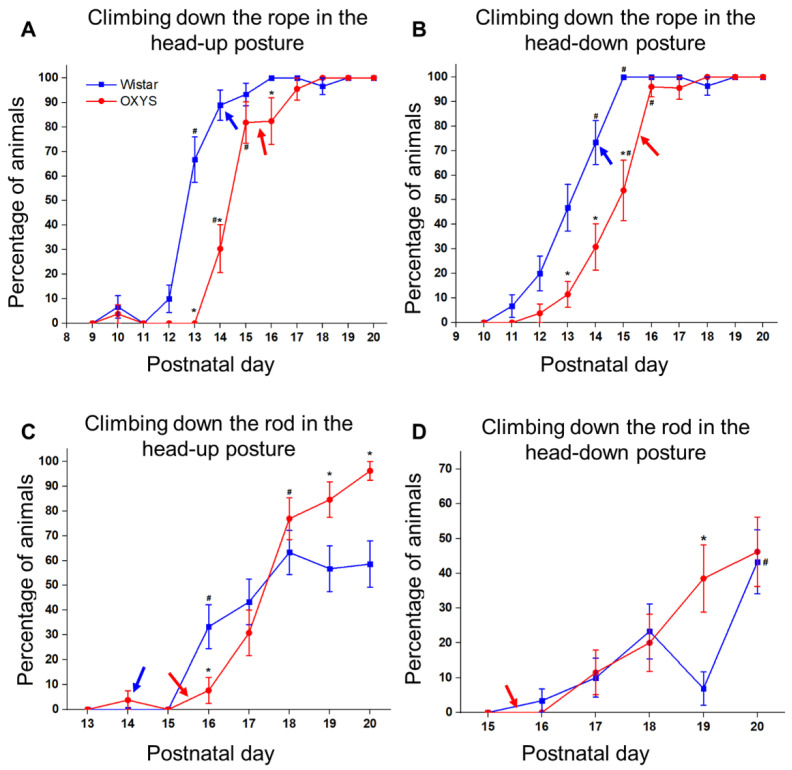
The abilities of OXYS and Wistar pups to climb down the rope and rod. (**A**) Climbing down the rope in the head-up posture; (**B**) climbing down the rope in the head-down posture; (**C**) climbing down the rod in the head-up posture; (**D**) climbing down the rod in the head-down posture. * *p* < 0.05 as compared to control (Wistar) rats; ^#^
*p* < 0.05 as compared to previous day; blue arrow: eye opening in Wistar pups; red arrow: eye opening in OXYS pups. The data are presented as the mean ± SEM.

**Figure 7 biomedicines-10-02910-f007:**
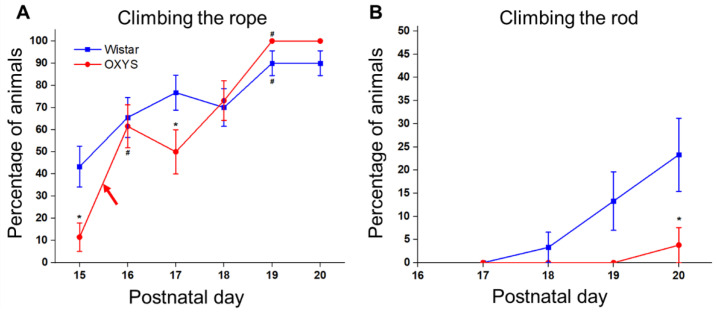
The abilities of OXYS and Wistar pups to climb the rope and rod. (**A**) Climbing the rope; (**B**) climbing the rod. * *p* < 0.05 as compared to control (Wistar) rats; ^#^
*p* < 0.05 as compared to a previous day; red arrow: eye opening in OXYS pups. The data are presented as the mean ± SEM.

**Figure 8 biomedicines-10-02910-f008:**
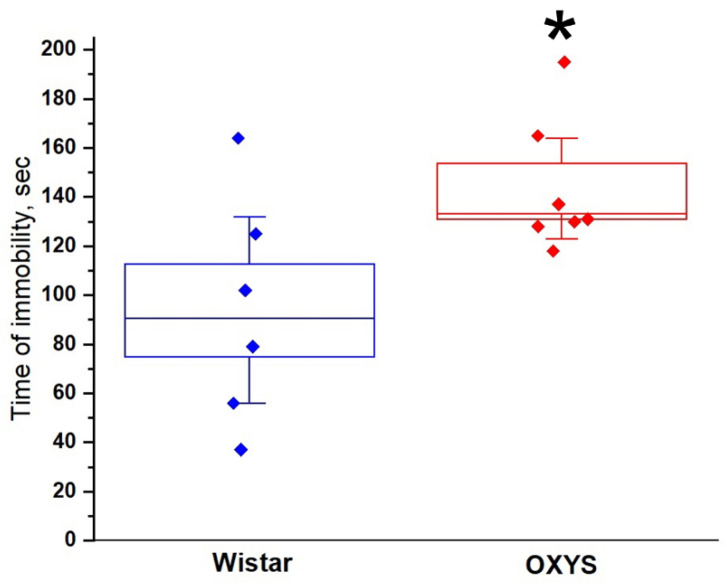
Duration of immobility in the forced swim test. * *p* < 0.05 as compared to control (Wistar) rats. The data are presented as the mean ± SEM.

**Table 1 biomedicines-10-02910-t001:** The scheme of the experiment.

Rat Strain	Sex	Age	Number of Animals	Measured Parameters
Wistar OXYS	Female	3 months	*n* = 8	Maternal behavior
Wistar OXYS	Male Female	PND1-PND20	Litters from 3 mothers of each strain (*n* = 62)	Raising of head Raising and coordination of limbs Locomotor activity Exploratory activity
Wistar OXYS	Male Female	PND10-PND20	Litters from 4 mothers of each strain (*n* = 80)	Righting in mid-air
Wistar OXYS	Male Female	PND1-PND10	Litters from 3 mothers of each strain (*n* = 66)	Placing reaction elicited by chin (“jumping”) Placing reaction elicited by vibrissae (“climbing”)
Wistar OXYS	Male Female	PND4-PND20	Litters from 3 mothers of each strain (*n* = 55)	Overcoming a gap between cages
Wistar OXYS	Male Female	PND5-PND20	Litters from 4 mothers of each strain (*n* = 80)	Jumping from height
Wistar OXYS	Male Female	PND3-PND20	Litters from 3 mothers of each strain (*n* = 56)	Ability to climb down (rope) Ability to climb down (rod) Ability to climb (rope) Ability to climb (rod)
Wistar OXYS	Male Female	PND12-PND15 PND16 PND17 PND18	4 siblings from 5 mothers of each strain (*n* = 40)	Habituation Novel object recognition (NOR) task Object location recognition (OLR) task Recency recognition (RR) task
Wistar OXYS	Male Female	PND45	*n* = 13	Depression-like behavior

**Table 2 biomedicines-10-02910-t002:** Manifestations of phenotypic parameters of OXYS and Wistar rat development.

Parameter	Literature Data [29]	Wistar			OXYS		
		♂	♀	♂ + ♀	♂	♀	♂ + ♀
Litter size	12	7.2 ± 0.24	7.3 ± 0.19	13.8 ± 0.15	5.8 ± 0.23 *	6.4 ± 0.21 *	11.2 ± 0.14 *
Auricle detachment	PND2	2.1 ± 0.03	2.1 ± 0.04	2.1 ± 0.02	3.1 ± 0.07 *	3.0 ± 0.06 *	3.1 ± 0.05 *
Emergence of pelage	PND5	5.0 ± 0.09	4.9 ± 0.08	5.0 ± 0.06	5.6 ± 0.08 *	5.5 ± 0.07 *	5.6 ± 0.05 *
Emergence of incisors	PND8	9.4 ± 0.18	9.0 ± 0.17	9.2 ± 0.13	9.7 ± 0.07	9.8 ± 0.08 *	9.7 ± 0.05 *
Eye opening	PND14	13.9 ± 0.10	13.8 ± 0.11	13.9 ± 0.08	15.6 ± 0.07 *	15.4 ± 0.07 *^,#^	15.5 ± 0.05 *
Descent of testes	PND25	24.1 ± 0.14	-	-	24.5 ± 0.19	-	-
Vaginal opening	PND30	-	29.6 ± 0.22	-	-	31.1 ± 0.25 *	-

* *p* < 0.05 as compared to control (Wistar) rats; ^#^ gender differences within a strain; PND: postnatal day. The data are presented as the mean ± SEM.

**Table 3 biomedicines-10-02910-t003:** Development of the quadruped stance and motion coordination.

Parameter	Wistar			OXYS		
	♂ (PND)	♀ (PND)	♂ + ♀ (PND)	♂ (PND)	♀ (PND)	♂ + ♀ (PND)
Raising of head	6.0 ± 0.38	6.25 ± 0.25	5.67 ± 0.20	6.77 ± 0.37	6.55 ± 0.28	6.68 ± 0.25 *
Raising of forelimbs	2.14 ± 0.34	2.75 ± 0.85	4.36 ± 0.33	3.47 ± 0.55	3.0 ± 0.66	3.29 ± 0.42 *
Raising of hindlimbs	2.29 ± 0.29	2.5 ± 0.65	2.45 ± 0.2	1.47 ± 0.3	1.55 ± 0.37	1.5 ± 0.23 *
Coordinated movements of forelimbs	4.14 ± 0.26	3.5 ± 0.65	5.15 ± 0.25	6.47 ± 0.38 *	6.82 ± 0.4 *	6.61 ± 0.28 *
Coordinated movements of hindlimbs	5.71 ± 0.64	6.75 ± 0.48	6.94 ± 0.23	7.18 ± 0.43	7.73 ± 0.47	7.39 ± 0.32

* *p* < 0.05 as compared to control (Wistar) rats; PND: postnatal day. The data are presented as the mean ± SEM.

**Table 4 biomedicines-10-02910-t004:** The average day of formation of the mid-air righting reflex and placing reactions.

Parameter	Wistar			OXYS		
	♂	♀	♂ + ♀	♂	♀	♂ + ♀
Mid-air righting	11.58 ± 0.41	12.11 ± 0.4	11.78 ± 0.3	12.15 ± 0.48	12.4 ± 0.54	12.23 ± 0.36 *
Placing reaction: jumping down before landing	4.95 ± 0.52	4.87 ± 0.56	4.9 ± 0.38	6.67 ± 0.51 *	5.22 ± 0.81	6.13 ± 0.46 (*p* = 0.05)
Placing reaction: climbing elevated object	1.9 ± 0.2	1.7 ± 0.17	1.79 ± 0.13	4.2 ± 0.44 *	4.67 ± 0.47 *	4.38 ± 0.32 *

* *p* < 0.05 as compared to control (Wistar) rats. The data are presented as the mean ± SEM.

**Table 5 biomedicines-10-02910-t005:** The average day of crossing the gap.

Parameter	Wistar			OXYS		
	♂	♀	♂ + ♀	♂	♀	♂ + ♀
Horizontal jumping 1.2 cm	8.35 ± 0.36	8.89 ± 0.54	8.6 ± 0.32	9.57 ± 0.81	8.7 ± 0.3	9.1 ± 0.38
Horizontal jumping 3.8 cm	18.09 ± 0.64	19.11 ± 0.26	18.55 ± 0.38	-	20.0 ± 0.0	20.0 ± 0 (*p* = 0.05)

The data are presented as the mean ± SEM.

**Table 6 biomedicines-10-02910-t006:** Assessment of the learning ability in Wistar and OXYS rats in the NOR test (PND16).

Parameter	Wistar			OXYS		
	♂	♀	♂ + ♀	♂	♀	♂ + ♀
Familiarization trial						
Distance, cm	117.00 ± 22.43	266.90 ± 69.88	191.95 ± 39.64	120.82 ± 34.90	87.33 ± 19.81	105.75 ± 21.01 *
Approaches to right object	0.20 ± 0.13	1.90 ± 1.06	1.05 ± 0.55	0.82 ± 0.38	0	0.45 ± 0.22
Time of investigation of right object, s	0.45 ± 0.32	4.45 ± 2.39	2.45 ± 1.26	3.64 ± 2.19	0	2.00 ± 1.25
Approaches to left object	0.30 ± 0.15	2.00 ± 0.89	1.15 ± 0.48	0.45 ± 0.28	0.33 ± 0.24	0.40 ± 0.18
Time of investigation of left object, s	1.60 ± 0.92	9.10 ± 4.80	5.35 ± 2.53	1.73 ± 1.44	0.67 ± 0.47	1.25 ± 0.81
Testing trial						
Distance, cm	214.60 ± 54.21	270.44 ± 76.72	241.05 ± 45.34	265.73 ± 66.39 ^$^	228.56 ± 54.74 ^$^	249.00 ± 43.14 ^$^
Approaches to familiar object	1.80 ± 0.65	2.33 ± 1.27	2.05 ± 0.67	3.18 ± 1.26	1.56 ± 0.73	2.45 ± 0.77 ^$^
Time of investigation of familiar object, s	11.20 ± 5.38	8.78 ± 4.19	10.05 ± 3.37	13.00 ± 6.88 ^$^	6.00 ± 3.07 ^$^	9.85 ± 4.02 ^$^
Approaches to novel object	1.80 ± 0.73	1.89 ± 1.07	1.84 ± 0.62	3.09 ± 1.37	1.78 ± 0.49	2.50 ± 0.78 ^$^
Time of investigation of novel object, s	5.30 ± 2.17	3.44 ± 1.89	4.42 ± 1.43	10.18 ± 5.28	6.11 ± 1.83 ^$^	8.35 ± 2.99 ^$^

* *p* < 0.05 as compared to control (Wistar) rats; ^$^
*p* < 0.05 for a difference between familiarization and test trials. The data are presented as the mean ± SEM.

**Table 7 biomedicines-10-02910-t007:** Assessment of the learning ability in Wistar and OXYS rats in the OLR test (PND17).

Parameter	Wistar			OXYS		
	♂	♀	♂ + ♀	♂	♀	♂ + ♀
Familiarization trial						
Distance, cm	137.50 ± 38.51	304.50 ± 51.57 ^#^	221.00 ± 36.71	98.64 ± 34.01	95.22 ± 24.59 *	97.10 ± 21.21 *
Approaches to right object	1.60 ± 0.75	3.70 ± 0.98	2.65 ± 0.65	1.18 ± 0.83	1.67 ± 0.71	1.40 ± 0.54
Time of investigation of right object, s	7.80 ± 2.96	11.30 ± 2.12	9.55 ± 1.82	4.18 ± 2.92	7.33 ± 4.15	5.60 ± 2.42
Approaches to left object	1.90 ± 0.97	4.00 ± 1.15	2.95 ± 0.77	1.09 ± 0.90	0.89 ± 0.39	1.00 ± 0.51
Time of investigation of left object, s	7.90 ± 4.77	14.60 ± 3.19	11.25 ± 2.90	6.18 ± 5.60	1.67 ± 0.73	4.15 ± 3.07
Testing trial						
Distance, cm	317.20 ± 62.23	306.00 ± 91.88	311.60 ± 54.02	269.55 ± 56.01	138.67 ± 230.72	210.65 ± 36.20
Approaches to familiar object	5.10 ± 1.40	4.10 ± 1.12	4.60 ± 0.88	4.45 ± 1.27 ^$^	2.00 ± 0.85	3.35 ± 0.83 ^$^
Time of investigation of familiar object, s	25.30 ± 7.86 ^$^	17.40 ± 6.49	21.35 ± 5.05 ^$^	18.09 ± 3.93	11.44 ± 6.98	15.10 ± 3.78 ^$^
Approaches to novel object	4.50 ± 1.28	4.40 ± 1.46	4.45 ± 0.95	5.82 ± 1.54 ^$^	1.67 ± 0.47	3.95 ± 0.98 ^$^
Time of investigation of novel object, s	17.20 ± 5.47 ^$^	19.50 ± 7.00	18.35 ± 4.33 ^$^	26.73 ± 7.25 ^$^	9.67 ± 5.33	19.05 ± 4.94 ^$^

* *p* < 0.05 as compared to control (Wistar) rats; ^#^
*p* < 0.05 as compared to male rats within the strain; ^$^
*p* < 0.05 for a difference between familiarization and test trials. The data are presented as the mean ± SEM.

**Table 8 biomedicines-10-02910-t008:** Assessment of the learning ability in Wistar and OXYS rats in the RR test (PND18).

Parameter	Wistar			OXYS		
	♂	♀	♂ + ♀	♂	♀	♂ + ♀
First familiarization trial						
Distance, cm	156.60 ± 40.32	304.90 ± 48.07	230.75 ± 34.95	63.36 ± 16.22	103.44 ± 24.27	81.40 ± 14.45
Approaches to right object	2.50 ± 1.24	5.70 ± 1.31	4.10 ± 0.95	0.36 ± 0.28	0.44 ± 0.24	0.40 ± 0.18
Time of investigation of the right object, s	11.50 ± 6.82	21.90 ± 5.96	16.70 ± 4.57	1.73 ± 1.38 *	1.00 ± 0.53 *	1.40 ± 0.78 *
Approaches to left object	2.30 ± 0.84	4.80 ± 1.06	3.55 ± 0.72	0.18 ± 0.12	0.67 ± 0.29	0.40 ± 0.15
Time of investigation of left object, s	21.10 ± 12.11	18.20 ± 4.81	19.65 ± 6.35	0.45 ± 0.37 *	3.00 ± 1.66 *	1.60 ± 0.80 *
Second familiarization trial						
Distance, cm	215.30 ± 30.54	236.90 ± 55.46	226.10 ± 30.91	56.00 ± 14.36	60.44 ± 18.72	58.00 ± 11.24
Approaches to right object	3.40 ± 1.00	4.00 ± 1.23	3.70 ± 0.77	0.18 ± 0.12	0.44 ± 0.24	0.30 ± 0.13
Time of investigation of right object, s	10.50 ± 2.87	12.50 ± 4.04	11.50 ± 2.42	0.73 ± 0.63	0.67 ± 0.37	0.70 ± 0.38
Approaches to left object	2.10 ± 0.81	3.50 ± 0.98	2.80 ± 0.64	0.18 ± 0.12	0.44 ± 0.34	0.30 ± 0.16
Time of investigation of left object, s	4.90 ± 1.72	13.90 ± 4.34	9.40 ± 2.49	0.27 ± 0.19	1.67 ± 1.18	0.90 ± 0.55
Testing trial						
Distance, cm	167.10 ± 27.85	127.20 ± 32.81	147.15 ± 21.44	75.00 ± 15.63 *	70.33 ± 16.11 *	72.90 ± 10.97 *
Approaches to familiar object	5.20 ± 1.30	2.10 ± 0.82	3.65 ± 0.83	1.27 ± 0.45 *	1.11 ± 0.65 *	1.20 ± 0.37 *
Time of investigation of familiar object, s	30.60 ± 10.54	13.20 ± 4.51	21.90 ± 5.92	5.73 ± 1.90 *	8.33 ± 4.58 *	6.90 ± 2.26 *
Approaches to novel object	3.70 ± 0.80	2.20 ± 1.06	2.95 ± 0.67	0.55 ± 0.28 *	0.89 ± 0.68 *	0.70 ± 0.33 *
Time of investigation of novel object, s	20.70 ± 9.23	5.80 ± 2.76	13.25 ± 4.99	3.36 ± 1.84	4.56 ± 3.40	3.90 ± 1.79

* *p* < 0.05 as compared to control (Wistar) rats. The data are presented as the mean ± SEM.

## Data Availability

Raw data are available from the corresponding author upon request.

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
