# Peer review of "Delayed Formation of Neonatal Reflexes and of Locomotor Skills Is Associated with Poor Maternal Behavior in OXYS Rats Prone to Alzheimer’s Disease-like Pathology"

_biomedicines, 2022, doi:10.3390/biomedicines10112910_

Round 1

Reviewer 1 Report

This is a significant study showing delayed formation of neonatal reflexes and locomotor skills associated with poor maternal behavior in OXYS rats prone to Alzheimer’s disease-like pathology. The study is well designed and performed, however, there are some concerns which should be addressed.

1. Randomization: Describe the procedure of the randomization method  or, if no randomization methods were used, then explicitly state in the manuscript “no randomization was performed”.

2. Blinding: Describe blinding procedures for each experiment (who was blinded to which procedure during the study, i.e. the experimenter was unaware of the animal’s group during experimentation, during (statistical) analysis, etc.). An experiment is also blind if the analysis or experimental group assignment is performed by a different person than the experimenter.

3. Predetermined sample size calculation: Specify if statistical methods were employed to predetermine the sample size and include a description of sample size calculations (provide all parameters of the calculation and estimation of effect size) in the manuscript or explicitly state in the Methods section of the manuscript that no sample calculation was performed.

4. State if any (and how many) animals were excluded based on the exclusion criteria or died during experiments.

5. Time-line diagram or flow-chart: Include in the Methods section a graphical time-line or flow-chart of the study design/ experimental procedure (including number of animals per group and number of excluded animals) and please provide a reference to the schematic in the manuscript text.

6. Describe in detail what exact measures were taken to minimize animal suffering after treatment/during experiments, and please specify if and what type of medication was given to reduce animal pain during experiments. A general statement such as, “All efforts were made to minimize animal suffering” is not sufficient. One must provide a detailed description of these “methods”.

7. Indicate the weight and housing (cage type, number of cage companions) of animals in the Methods

8. Describe in a separate section what statistical analyses were carried out, their rationale, power analyses. Specify if an assessment (and which one) of the normality of data was carried out. Specify if a test for outliers (and which one) was conducted on the data and state if any data points were excluded.

9. Show individual data points (mandatory for small sample sizes n<15) as a dotplot or use box-plots instead of simple bar graphs. More information as to why this is important can be found here:

https://journals.plos.org/plosbiology/article?id=10.1371/journal.pbio.1002128

10. The title of the manuscript sounds rather awkward. I would suggest to substitute "against the Background of" for "associated with" or even "is associated with".

Author Response

This is a significant study showing delayed formation of neonatal reflexes and locomotor skills associated with poor maternal behavior in OXYS rats prone to Alzheimer’s disease-like pathology. The study is well designed and performed, however, there are some concerns which should be addressed.

We are grateful to the reviewer for the positive evaluation of our study!

1. Randomization: Describe the procedure of the randomization method or, if no randomization methods were used, then explicitly state in the manuscript “no randomization was performed”.

Thank you for the comment; no randomization was performed.

2. Blinding: Describe blinding procedures for each experiment (who was blinded to which procedure during the study, i.e. the experimenter was unaware of the animal’s group during experimentation, during (statistical) analysis, etc.). An experiment is also blind if the analysis or experimental group assignment is performed by a different person than the experimenter.

Thanks for the comment; we added this information into the Materials and Methods section.

3. Predetermined sample size calculation: Specify if statistical methods were employed to predetermine the sample size and include a description of sample size calculations (provide all parameters of the calculation and estimation of effect size) in the manuscript or explicitly state in the Methods section of the manuscript that no sample calculation was performed.

We clarified this issue in the manuscript.

4. State if any (and how many) animals were excluded based on the exclusion criteria or died during experiments.

We added this information into the Materials and Methods section.

5. Time-line diagram or flow-chart: Include in the Methods section a graphical time-line or flow-chart of the study design/ experimental procedure (including number of animals per group and number of excluded animals) and please provide a reference to the schematic in the manuscript text.

Thanks for the suggestion; we added the diagram.

6. Describe in detail what exact measures were taken to minimize animal suffering after treatment/during experiments, and please specify if and what type of medication was given to reduce animal pain during experiments. A general statement such as, “All efforts were made to minimize animal suffering” is not sufficient. One must provide a detailed description of these “methods”.

We inserted this important information into the Materials and Methods section.

7. Indicate the weight and housing (cage type, number of cage companions) of animals in the Methods

We now indicate these parameters in the Materials and Methods section.

8. Describe in a separate section what statistical analyses were carried out, their rationale, power analyses. Specify if an assessment (and which one) of the normality of data was carried out. Specify if a test for outliers (and which one) was conducted on the data and state if any data points were excluded.

Thank you for noticing; we added this information into the Materials and Methods section (subsection 2.4. Statistics).

9. Show individual data points (mandatory for small sample sizes n<15) as a dotplot or use box-plots instead of simple bar graphs. More information as to why this is important can be found here:

https://journals.plos.org/plosbiology/article?id=10.1371/journal.pbio.1002128

We redid the graphs and added individual data points. However, because of summarizing results from both sexes, almost all graphs contained more than 15 individual data points for every time point.

10. The title of the manuscript sounds rather awkward. I would suggest to substitute "against the Background of" for "associated with" or even "is associated with".

We agree; we revised this expression in the title.

Reviewer 2 Report

It is a very interesting manuscript demonstrating the differences between OXYS and Wistar rats and the potential role of maternal care in developing depressive-like behavior. I have only one general remark – it is necessary to describe more in detail the number of animals used in each phase.

The manuscript need English language and style corrections.

Author Response

It is a very interesting manuscript demonstrating the differences between OXYS and Wistar rats and the potential role of maternal care in developing depressive-like behavior. I have only one general remark – it is necessary to describe more in detail the number of animals used in each phase.
Thank you very much for the praise of our manuscript! We added the diagram with experimental procedures and the number of animals in the Materials and Methods section.

The manuscript need English language and style corrections.

Sorry for this oversight; we hired an editing company to improve the English language.

Round 2

Reviewer 1 Report

Ноу authors have addressed all concerns.